# ESTIMATING HETEROGENEOUS TREATMENT EFFECT WITH DELAYED RESPONSE

## ABSTRACT

Estimation of heterogeneous treatment effects has gathered much attention in recent years and has been widely adopted in medicine, economics, and marketing. Previous studies assumed that one of the potential outcomes of interest could be observed timely and accurately. However, a more practical scenario is that treatment takes time to produce causal effects on the outcomes. For example, drugs take time to produce medical utility for patients and users take time to purchase items after being recommended, and ignoring such delays in feedback can lead to biased estimates of heterogeneous treatment effects. To address the above problem, we study the impact of observation time on estimating heterogeneous treatment effects by further considering the potential response time that potential outcomes have. We theoretically prove the identifiability results and further propose a principled learning approach, known as CFR-DF (Counterfactual Regression with Delayed Feedback), to simultaneously learn potential response times and potential outcomes of interest. Results on both simulated and real-world datasets demonstrate the effectiveness of our method.

## 1 INTRODUCTION

Heterogeneous treatment effects (HTE) estimation using observational data is a fundamental problem that applies to a wide variety of areas (Alaa & Van Der Schaar, 2017; Alaa et al., 2017; Hannart et al., 2016; LaLonde, 1986; Shalit et al., 2017). For example, in precision medicine, physicians decide drug allocation by the treatment effect of the patient on the drug (Jaskowski & Jaroszewicz, 2012). In online markets, the causal effect of recommending an item on a user's purchase behavior is used for personalized recommendations (Schnabel et al., 2016). Unlike using observed outcomes to make decisions, HTE accounts for variations in both factual outcomes and counterfactual outcomes among individuals or subgroups. The challenge lies in accurately estimating and learning HTE due to the unobserved nature of the counterfactual outcomes with alternative treatment (Holland, 1986).

Many methods have been proposed to estimate HTE from observational data. For instance, representation learning-based approaches learn a covariate representation that is independent of the treatment to overcome the covariate shift between the treatment and control groups (Johansson et al., 2016; Shalit et al., 2017; Shi et al., 2019; Yao et al., 2018). The tree-based approach generalizes Bayesian inference and random forest methods for nonparametric estimation (Chipman et al., 2010; Wager & Athey, 2018). The generative model-based approaches use the widely adopted variational autoencoder and generative adversarial network to generate individual counterfactual outcomes (Louizos et al., 2017; Yoon et al., 2018). These studies have also been extended to continuous treatment scenarios (Bica et al., 2020; Nie et al., 2021; Schwab et al., 2018; 2020).

Existing methods require that one of the potential outcomes of interest could be observed timely and accurate. However, interventions on individuals usually do not affect outcomes of interest immediately, and treatment takes time to produce causal effects on the outcomes. For example, drugs take time to produce medical utility for patients, with the long-term prognosis as the outcome of interest, which benefits the treatment decision from the physicians. In online markets, a recommendation algorithm focuses on whether or not the user will eventually purchase, but users take time to purchase items after being recommended (Chapelle, 2014), which poses a critical challenge in practice: as in Figure 1(a), if the observation window is too short, some samples will be incorrectly marked as negative whose conversion will occur in the future; but if it is too long, the recommendation algorithm will not

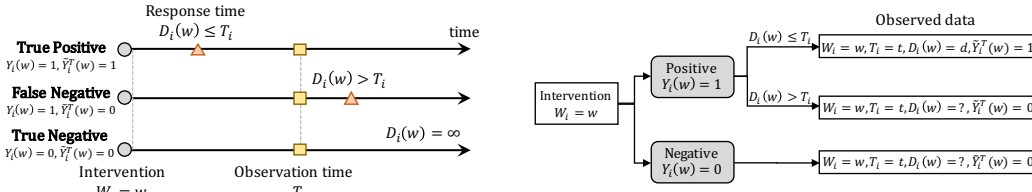

(a) Three types of delayed response scenarios.

(b) Observed data with various potential outcomes.

Figure 1: Illustrations for false negative (left) and observed data format (right) under delayed response.

be able to guarantee its timely availability (Yoshikawa & Imai, 2018). In summary, ignoring such delays in outcome response can lead to biased estimates of heterogeneous treatment effects.

In this paper, we first formalize the HTE estimation problem in the presence of delayed response. In contrast to previous studies that only considered the effect of treatment on outcome, we also consider potential response times with different treatments, since treatment may affect response time, e.g., users who receive item recommendations purchase more quickly. Therefore, as in Figure 1(a), given the treatment $w$ for an individual, even the eventual outcome of interest $Y(w)$ is positive, e.g., the user will eventually purchase the item, we can only observe the true positive conversion ($Y(w) = 1, \tilde{Y}(w) = 1$) when the potential response time is less than the observation time ($D(w) \leq T$), while observing the false negative outcome ($Y(w) = 1, \tilde{Y}(w) = 0$) vise versa. Instead, when the eventual outcome $Y(w)$ is negative, e.g., the user never purchases the item, then we observe the negative outcome ($\tilde{Y}(w) = 0$) regardless of the observation time. Figure 1(b) illustrates the format of the observed data, which comes with an additional challenge, that is, we could not obtain the exact value of the response time if the positive feedback did not occur before the observation time.

To address the above issues, we study the impact of observation time on estimating heterogeneous treatment effects by further considering the potential response time that potential outcomes have. Theoretically, we prove the eventual potential outcomes are identifiable in the whole population, which is essential for treatment allocation. For subgroups in which individuals always have positive eventual outcomes regardless of treatment, we also show the identifiability of potential response times, which quantifies the causal effect of treatment on response times. Using the eventual outcomes as hidden variables, we reconstruct the posterior distribution of a delayed response and provide explicit solutions to estimate the parameters of interest within a modified EM algorithm. Furthermore, we propose a principled learning approach that extends counterfactual regression (CFR) to delayed feedback outcomes, named CFR-DF, to simultaneously predict potential outcomes and potential response times. Finally, we discuss the importance of this work for policy learning and validate the effectiveness of the proposed method on both synthetic and real-world datasets.

The main contributions of this paper are summarized as follows:

- We formalize the HTE estimation problem with delayed response, in which treatment takes time to produce a causal effect on the outcome.
- We theoretically prove the eventual potential outcome is identifiable in the whole population, and show the identifiability of potential response times on the always-positive stratum.
- We propose a principled learning algorithm, called CFR-DF, that utilizes the EM algorithm to estimate both eventual potential outcomes and potential response times.
- We perform extensive experiments on both synthetic and real-world datasets to show the effectiveness of the proposed approach in estimating HTE with delayed responses.

## 2 RELATED WORK

In Heterogeneous treatment effect (HTE) estimation, non-random treatment assignments can result in different probabilities of missing covariates in different treatment arms, which may introduce confounding bias. To address this issue, most methods strive to balance covariates to estimate HTE accurately, such as matching, stratification, outcome regression, weighting, and doubly robust methods (Rosenbaum, 1987; Rosenbaum & Rubin, 1983; Li et al., 2016; Hainmueller, 2012). With

the advances in deep learning, Balancing Neural Network (BNN) (Johansson et al., 2016) and CounterFactual Regression (CFR) (Shalit et al., 2017) propose to learn a covariate representation that is independent of the treatment to overcome the covariate shift between the treatment and control groups, in which the independence is measured by Integral Probability Metric (IPM) (Johansson et al., 2016; Shalit et al., 2017). SITE (Yao et al., 2018) preserves local similarity and balances the distributions of the representation simultaneously. Motivated by targeted learning (van der Laan & Rose, 2011), DragonNet (Shi et al., 2019) proposed an adaptive neural network to end-to-end model propensity scores and counterfactual outcomes. DR-CFR (Hassanpour & Greiner, 2020) and DeR-CFR (Wu et al., 2022) propose a disentanglement framework to identify the representation of confounders from all observed variables. By exploiting the generative models, CEVAE (Louizos et al., 2017) and GANITE (Yoon et al., 2018) generate counterfactual outcomes for HTE estimation. However, these algorithms rely on timely and accurate observation of the eventual potential outcomes.

In practice, interventions usually take time to have a causal effect on the outcome (Chapelle, 2014; Yoshikawa & Imai, 2018). Despite the problem setup and the causal estimand of interest is different, many studies have examined HTE estimation under time-to-event data. Curth et al. (2021) used neural networks for discrete time analyses and Chapfuwa et al. (2021) used generative models for counterfactual time-to-event data analysis in continuous time. Based on the Cox model (Cox, 1972), Schrod et al. (2022) proposed a treatment-specific semi-parametric Cox loss using time-to-event data for treatment optimization. Gupta et al. (2023) derived a binary treatment evidence lower bound (ELBO) for parametric survival analysis, and designed a neural network for learning the per-individual survival density. Different from Chapfuwa et al. (2021); Curth et al. (2021), Curth & van der Schaar (2023) considered time-to-event data with competing events, which can act as an additional source of covariate shift. In addition, Nagpal et al. (2022) presented a latent variable approach to mediate the base survival rates and help determine the effects of an intervention. Nagpal et al. (2023) extended Nagpal et al. (2022) by proposing a statistical approach to recovering sparse phenogroups (or subtypes) that demonstrate differential treatment effects as compared to the study population. Though delayed response can be considered as a right-censored problem, rather than focusing on the effect of treatment on survival curves, this paper assumes that it takes time to yield an observable outcome that eventually has a positive outcome (e.g., conversion in uplift modeling) and considers both conversion time and whether or not to convert as potential outcomes by utilizing a *hybrid model*. By considering the joint potential outcome of individuals from a principal stratification perspective (Frangakis & Rubin, 2002; Pearl, 2011), we theoretically prove that the potential response times on subgroups in which individuals always have positive eventual outcomes regardless of treatment are identifiable. It is also interesting to note that the problem studied in this paper can also be considered as a noisy label on the eventual outcome of interest due to the limited observation time, which causes the previous HTE methods to be biased.

## 3 HETEROGENEOUS TREATMENT EFFECT WITH DELAYED RESPONSE

### 3.1 NOTATION AND SETUP

In this paper, we consider the case of binary treatment. Suppose a simple random sample of $n$ units from a super population $\mathbb{P}$, for each unit $i$, the covariate and the assigned treatment are denoted as $X_i \in \mathcal{X} \subset \mathbb{R}^m$ and $W_i \in \mathcal{W} = \{0, 1\}$, where $W_i = 1$ means receiving the treatment and $W_i = 0$ means not receiving the treatment, respectively. Different from the previous problem setup in both standard HTE estimation (Johansson et al., 2016; Shalit et al., 2017; Shi et al., 2019; Yao et al., 2018) and recent time-to-event studies related to survival analysis (Gupta et al., 2023; Chapfuwa et al., 2021; Curth et al., 2021), we consider the response time from the imposing treatment to producing influence on the outcome. Specifically, let $Y_i \in \mathcal{Y} = \{0, 1\}$ be the binary outcome at the eventual time, e.g., whether a user will eventually purchase, as the primary outcome of interest, and we call unit with $Y_i = 1$ as a positive sample. Without loss of generality, the time at which the treatment $W_i$ is imposed on unit $i$ is taken as the start time, let $D_i$ be the response time for individuals with $Y_i = 1$ to produce positive feedback, and we set $D_i = \infty$ for individuals with $Y_i = 0$. As shown in Figure 1(a), given an observation time $T_i$, we see a positive feedback at $T_i$, denoted as $\tilde{Y}_i^T = 1$, if and only if individual $i$ is a positive sample $Y_i = 1$ with the response time $D_i \leq T_i$, and marked as *true positive*. However, for some other positive samples with $Y_i = 1$, we would see false negative feedback $\tilde{Y}_i^T = 0$ at the observation time $T_i$, when the response time is greater than the observation

Table 1: The units are divided into four strata based on the joint potential outcomes $(Y(0), Y(1))$.

| GROUP | $Y(0)$ | $Y(1)$ | $D(0)$ | $D(1)$ | DESCRIPTION | PREFERRED TREATMENT |
|-------|--------|--------|--------|--------|-------------|---------------------|
| PP | 1 | 1 | ✓ | ✓ | ALWAYS-POSITIVE | DEPENDS ON $\tau_D(x)$ |
| NP | 0 | 1 | $\infty$ | ✓ | USEFUL TREATMENT | TREATMENT ($W = 1$) |
| PN | 1 | 0 | ✓ | $\infty$ | HARMFUL TREATMENT | CONTROL ($W = 0$) |
| NN | 0 | 0 | $\infty$ | $\infty$ | ALWAYS-NEGATIVE | EITHER ($W = 0$ OR 1) |

time, i.e., $D_i > T_i$, and marked as *false negative*. For samples that never yield positive outcomes, we observe negative feedback $\tilde{Y}_i^T = 0$ for all observation times $T_i$, and marked as *true negative*.

To study the effect of treatment on the eventual outcome and the response time, we adopt the potential outcome framework (Rubin, 1974; Neyman, 1990) in causal inference. Specifically, let $Y_i(0)$ and $Y_i(1)$ be the eventual outcome of unit $i$ had this unit receive treatment $W_i = 0$ and $W_i = 1$, respectively. In addition, since treatment may have an effect on the response time, e.g., users purchase more quickly when receiving ads about an item, we denote $D_i(0)$ and $D_i(1)$ be the potential response time had unit $i$ receive treatment $W_i = 0$ and $W_i = 1$, respectively. Therefore, given an observation time $T_i$, the corresponding potential outcomes $\tilde{Y}_i^T(0)$ and $\tilde{Y}_i^T(1)$ can be analogously defined. Since each unit can be only assigned with one treatment, we always observe the corresponding outcome to be either $\tilde{Y}_i^T(0)$ or $\tilde{Y}_i^T(1)$, but not both, which is also known as the fundamental problem of causal inference (Holland, 1986; Morgan & Winship, 2015). However, one should note that similar conclusions no longer hold for the eventual potential outcomes $(Y_i(0), Y_i(1))$ and the potential response times $(D_i(0), D_i(1))$, as we cannot observe the exact eventual outcome as well as the response time due to the limited observation time.

We assume that the observation for unit $i$ is $\tilde{Y}_i^T = (1 - W_i)\tilde{Y}_i^T(0) + W_i\tilde{Y}_i^T(1)$. In other words, the observed outcome at time $T_i$ is the potential outcome corresponding to the assigned treatment, which is also known as the consistency assumption in the causal literature. We assume that the stable unit treatment value assumption (STUVA) assumption holds, i.e., there should not be alternative forms of treatment and interference between units. Furthermore, we assume the positivity of treatment assignment, i.e., $\eta < \mathbb{P}(W_i = 1 | X_i = x) < 1 - \eta$, where $\eta$ is a constant between 0 and $1/2$.

We summarize the observed data formats in Figure 1(b), with the following three cases.

- True positive ($Y_i(w) = 1, \tilde{Y}_i^T(w) = 1$) with observed ($W_i = w, D_i(w) = d \leq T_i, \tilde{Y}_i^T(w) = 1$);
- False negative ($Y_i(w) = 1, \tilde{Y}_i^T(w) = 0$) with observed ($W_i = w, T_i = t, \tilde{Y}_i^T(w) = 0$);
- True negative ($Y_i(w) = 0, \tilde{Y}_i^T(w) = 0$) with observed ($W_i = w, T_i = t, \tilde{Y}_i^T(w) = 0$),

which leads to an additional challenge due to one cannot distinguish between *false negative* and *true negative* directly from the observed data ($W_i = w, T_i = t, \tilde{Y}_i^T(w) = 0$).

### 3.2 PARAMETERS OF INTEREST

We consider two meaningful parameters of interest in the following. For simplification, we drop the subscript $i$ for a generic unit hereafter. First, unlike previous studies that focused on the HTE of treatment on current observed outcomes, i.e., $\tau^T(x) = \mathbb{E}[\tilde{Y}^T(1) - \tilde{Y}^T(0) \mid X = x]$, we focused on the HTE of treatment on the eventual outcomes, i.e., $\tau(x) = \mathbb{E}[Y(1) - Y(0) \mid X = x]$. Notably, the latter poses two challenges: first, the confounding bias introduced by covariates, which is similar to previous studies; second, how to recover the eventual outcome $Y$ of interest from the observed outcome $\tilde{Y}^T$ at time $T$. When the observation time $T$ is sufficiently long to exceed the response time $D$ for all individuals, the proposed causal estimand $\tau(x)$ degenerates to $\tau^T(x)$.

Next, we show that individuals can be divided into four strata by considering the joint potential outcomes $(Y(0), Y(1))$, as shown in Table 1, and named as the *always-positive* stratum, *useful treatment* stratum, *harmful treatment* stratum, and *always-negative* stratum accordingly. From a policy learning perspective, it is clear that treatment should be given and not given to individuals in *useful treatment* stratum and *harmful treatment* stratum, respectively. For individuals in the *always-negative* stratum, for example, users who will never purchase or patients who will always be cured regardless of treatment, either of the treatments is reasonable and results in no difference. When

considering individuals in the *always-positive* stratum, despite having both $Y(0) = 1$ and $Y(1) = 1$ for the eventual outcomes, it is meaningful to study the HTE of the treatment on the response times. Formally, the causal estimand of interest is $\mathbb{E}[D(1) - D(0) \mid Y(0) = 1, Y(1) = 1, X = x]$. For the other three strata, since there exists a treatment $w$ such that $Y(w) = 0$, the corresponding response time can be regarded as $D(w) = \infty$, resulting in HTE of treatment on response time being ill-defined.

We summarize the causal estimand of interest as follows.

- HTE on the eventual outcome: $\tau(x) = \mathbb{E}[Y(1) - Y(0) \mid X = x]$;
- HTE on the response time: $\tau_D(x) = \mathbb{E}[D(1) - D(0) \mid Y(0) = 1, Y(1) = 1, X = x]$.

### 3.3 IDENTIFIABILITY RESULTS

We then discuss the identifiability of the causal parameters of interest in Section 3.2. We adopt and refer to the following assumptions.

**Assumption 1** (Unconfoundedness). $W \perp\!\!\!\perp (D(0), D(1), \tilde{Y}^t(0), \tilde{Y}^t(1)) \mid X$ *for all* $t > 0$.

**Assumption 2** (Time Independence). $T \perp\!\!\!\perp (D(0), D(1), \tilde{Y}^t(0), \tilde{Y}^t(1), W) \mid X$ *for all* $t > 0$.

**Assumption 3** (Time Sufficiency). $\inf\{d : F_D^{(w)}(d \mid Y(w) = 1, X) = 1\} < \inf\{t : F_T(t) = 1\}$ *for* $w = 0, 1$, *where* $F(\cdot)$ *is the cumulative distribution function (cdf).*

**Assumption 4** (Monotonicity). $Y(0) \leq Y(1)$.

**Assumption 5** (Principal Ignorability). $(W, Y(w)) \perp\!\!\!\perp D(1 - w) \mid Y(1 - w), X$ *for* $w = 0, 1$.

Among them, unconfoundedness is also known as no unmeasured confounders assumption as it holds if all variables that affect both treatment and potential outcomes are included in $X$. Time independence holds since the observation occurs after the treatment, and the observation does not affect the potential response times $D(w)$ and the potential outcomes $\tilde{Y}^t(w)$ at a given time $t > 0$ for $w = 0, 1$. Time Sufficiency means that we need a subset of individuals (not all) with observed outcomes $\tilde{Y} = 1$ to identify eventual potential outcomes, which is a necessary condition for studying survival analysis. Monotonicity assumption is plausible in many applications when the effect of the decision on the outcome is non-negative for all individuals, e.g., drug is not harmful to the patient or recommendations do not have a negative effect on user purchases. Principal Ignorability requires that the expectations of the potential outcomes do not vary across principal strata conditional on the covariates. It is widely used in applied statistics (Imai & Jiang, 2020; Ben-Michael et al., 2022).

We next provide the identifiability results of three causal parameters (see Appendix A.1 for proofs).

**Theorem 1.** *Under Assumptions 1-3, the HTE on the eventual outcome $\tau(x)$ is identifiable.*

To proceed to identify the HTE of treatment on potential response times in the *always-positive* stratum, we introduce monotonicity assumption to identify the probability of belonging to this stratum.

**Lemma 1.** *Under Assumptions 1-4, $\mathbb{P}(Y(0) = 1, Y(1) = 1 \mid X = x)$ is identifiable.*

Following the previous studies (Imai & Jiang, 2020; Ben-Michael et al., 2022; Jiang et al., 2022), we assume principal ignorability holds to identify the HTE of treatment on potential response times in the *always-positive* stratum. Under all of the above assumptions, $\tau_D(x)$ is also identifiable.

**Theorem 2.** *Under Assumptions 1-5, the HTE on the response time in the always-positive stratum $\tau_D(x) = \mathbb{E}[D(1) - D(0) \mid Y(0) = 1, Y(1) = 1, X = x]$ is identifiable.*

## 4 CFR-DF: COUNTERFACTUAL REGRESSION WITH DELAYED FEEDBACK

In this section, we propose a principled learning approach to perform **C**ounter**F**actual **R**egression with **D**elayed **F**eedback on outcomes, named CFR-DF. Specifically, CFR-DF consists of two sets of models to predict the eventual potential outcomes, i.e., $\mathbb{P}(Y(0) = 1 \mid X = x)$ and $\mathbb{P}(Y(1) = 1 \mid X = x)$ and the potential response times, i.e., $\mathbb{P}(D(0) = d \mid X = x, Y(0) = 1)$ and $\mathbb{P}(D(1) = d \mid X = x, Y(1) = 1)$, respectively, the former of which can be flexibly exploited from previous HTE estimation methods in the following framework, and we take the widely used counterfactual regression (CFR) (Shalit et al., 2017) for illustration purpose.

Recall that in Figure 1(b), we show two possible observed data formats. On the one hand, the probability of observing positive feedback $\tilde{Y}^T = 1$ with response time $D = d$ at time $T = t > d$:

$$
\mathbb{P}(\tilde{Y}^T = 1, D = d \mid X = x, W = w, T = t) = \mathbb{P}(Y = 1, D = d \mid X = x, W = w)
$$
$$
= \mathbb{P}(Y(w) = 1 \mid X = x, W = w)\mathbb{P}(D(w) = d \mid X = x, W = w, Y(w) = 1)
$$
$$
= \mathbb{P}(Y(w) = 1 \mid X = x)\mathbb{P}(D(w) = d \mid X = x, Y(w) = 1), \tag{1}
$$

where the first equality follows from time independence, the second equality follows from the consistency assumption, and the last equality follows from the unconfoundedness assumption.

On the other hand, by the law of total probabilities, and again using the conditional independence of observation time, the probability of not having observed positive feedback at time $T = t > d$ is:

$$
\mathbb{P}(\tilde{Y}^T = 0 \mid X = x, W = w, T = t) = \mathbb{P}(Y = 0 \mid X = x, W = w)\mathbb{P}(\tilde{Y}^t = 0 \mid X = x, W = w, Y = 0)
$$
$$
+ \mathbb{P}(Y = 1 \mid X = x, W = w)\mathbb{P}(\tilde{Y}^t = 0 \mid X = x, W = w, Y = 1), \tag{2}
$$

where $\mathbb{P}(Y = 0 \mid X = x, W = w)$ is equivalent to $\mathbb{P}(Y(w) = 0 \mid X = x)$ by unconfoundedness assumption, with similar result holds for $\mathbb{P}(Y = 1 \mid X = x, W = w)$. In addition, we have $\mathbb{P}(\tilde{Y}^t = 0 \mid X = x, W = w, Y = 0) = 1$, due to eventual outcome $Y = 0$ implies $\tilde{Y}^t = 0$ for all $t > 0$. Next we focus on the last item $\mathbb{P}(\tilde{Y}^t = 0 \mid X = x, W = w, Y = 1)$.

By noting the equivalence between $(\tilde{Y}^t(w) = 0, Y(w) = 1)$ and $(D(w) > t, Y(w) = 1)$, we have:

$$
\mathbb{P}(\tilde{Y}^t = 0 \mid X = x, W = w, Y = 1) = \mathbb{P}(\tilde{Y}^T(w) = 0 \mid X = x, Y(w) = 1, T = t)
$$
$$
= \mathbb{P}(D(w) > t \mid X = x, Y(w) = 1) = \int_t^\infty \mathbb{P}(D(w) = u \mid X = x, Y(w) = 1)du. \tag{3}
$$

With the above results, we have the probability of $\tilde{Y}^T = 0$ at time $T = t$ is:

$$
\mathbb{P}(\tilde{Y}^T = 0 \mid X = x, W = w, T = t)
$$
$$
= \mathbb{P}(Y(w) = 0 \mid X = x) + \mathbb{P}(Y(w) = 1 \mid X = x) \int_t^\infty \mathbb{P}(D(w) = u \mid X = x, Y(w) = 1)du, \tag{4}
$$

which can be represented by two sets of models in CFR-DF.

Different from CFR, an essential challenge is that we cannot observe the eventual outcomes $Y$, which results in the unavailability to directly fit the potential outcomes of interest $\mathbb{P}(Y(w) = 0 \mid X = x)$ and $\mathbb{P}(Y(w) = 1 \mid X = x)$ from the observed data. To address this problem, we treat the eventual potential outcomes as latent variables, and estimate the parameters of interest using a modified EM algorithm as below, which addresses both the confounding bias and the missing eventual outcomes.

**Expectation Step.** For a given data point $(x_i, w_i, t_i, y_i^t)$, we need to compute the posterior probability of the hidden variable $p_i := \mathbb{P}(Y_i(w_i) = 1 \mid X = x_i, W = w_i, T = t_i, \tilde{Y}^T = y_i^t)$. If positive feedback $y_i^t = 1$ is observed at time $T = t$, then it is obvious that $p_i = 1$ for unit $i$. Alternatively, if $y_i^t = 0$ is observed at time $t$ for individual $i$, then the posterior probability $p_i$ can be expressed as:

$$
p_i = \mathbb{P}(Y_i(w_i) = 1 \mid X = x_i, W = w_i, T = t_i, \tilde{Y}_i^T = 0)
$$
$$
= \frac{\mathbb{P}(\tilde{Y}_i^T(w_i) = 0 \mid X = x_i, Y_i(w_i) = 1, T = t_i)\mathbb{P}(Y_i(w_i) = 1 \mid X = x_i)}{\mathbb{P}(\tilde{Y}_i^T = 0 \mid X = x_i, W = w_i, T = t_i)} \tag{5}
$$
$$
= \frac{\mathbb{P}(Y_i(w_i) = 1 \mid X = x_i) \int_{t_i}^\infty \mathbb{P}(D_i(w_i) = u \mid X = x_i, Y_i(w_i) = 1)du}{\mathbb{P}(Y_i(w_i) = 0 \mid X = x_i) + \mathbb{P}(Y_i(w_i) = 1 \mid X = x_i) \int_{t_i}^\infty \mathbb{P}(D_i(w_i) = u \mid X = x_i, Y_i(w_i) = 1)du},
$$

which can be calculated from the maximization step of the models in CFR-DR in the following.

**Maximization Step.** Given the hidden variable values $p_i$ computed from the E step, let $S = s_i$ denote $(X = x_i, W = w_i, T = t_i)$, we maximize the expected log-likelihood during the M step:

$$
\sum_{i:\tilde{y}_i^t = 1} \log \mathbb{P}(\tilde{Y}_i^T = 1, D = d_i \mid S = s_i) + \sum_{i:\tilde{y}_i^t = 0}(1 - p_i) \log \mathbb{P}(\tilde{Y}_i^T = 0, Y_i(w_i) = 0 \mid S = s_i)
$$
$$
+ \sum_{i:\tilde{y}_i^t = 0} p_i \log \mathbb{P}(\tilde{Y}_i^T = 0, Y_i(w_i) = 1 \mid S = s_i). \tag{6}
$$

From a similar argument as derived above, the expected log-likelihood is equal to:

$$
\sum_i p_i \log \mathbb{P}(Y_i(w_i) = 1 \mid X = x_i) + (1 - p_i) \log(1 - \mathbb{P}(Y_i(w_i) = 1 \mid X = x_i)) \tag{7}
$$
$$
+ \sum_{i:\tilde{y}_i^t = 1} \log \mathbb{P}(D_i(w_i) = d_i \mid X = x_i, Y_i(w_i) = 1) + \sum_{i:\tilde{y}_i^t = 0} p_i \log \int_{t_i}^\infty \mathbb{P}(D(w_i) = u \mid X = x_i, Y_i(w_i) = 1)du,
$$

where *the eventual potential outcome model* $\mathbb{P}(Y(w) = 1 \mid X = x)$ *and the potential response time model* $\mathbb{P}(D(w) = d \mid X = x, Y(w) = 1)$ *can be optimized independently*. Due to space limitations, the computation details of parametric and non-parametric EM models are deferred to Appendix A.2.

Let $h^Y(\Phi^Y(x), w)$ be the prediction model for the eventual potential outcomes $\mathbb{P}(Y(w) = 1 \mid X = x)$, and $h^D(\Phi^D(x), w, d)$ be the prediction model for the potential response times $\mathbb{P}(D(w) = d \mid X = x, Y(w) = 1)$, where $\Phi^Y : \mathcal{X} \to \mathcal{R}^Y$ and $\Phi^D : \mathcal{X} \to \mathcal{R}^D$ are the covariate representations, $\mathcal{R}^Y$ and $\mathcal{R}^D$ are the representation spaces, and $h^Y : \mathcal{R}^Y \times \{0, 1\} \to \mathcal{Y}$ and $h^D : \mathcal{R}^D \times \{0, 1\} \times \mathbb{R}^+ \to \mathbb{R}^+$ are the prediction heads, respectively. Inspired by CFR (Shalit et al., 2017), we take the Integral Probability Metric (IPM) distance induced by the representations as a penalty term, to control the generalization error caused by covariate shift between the treatment and control group.

Given the posterior probabilities $p_i$ computed from the E step, we train the eventual potential outcome model by minimizing the derived negative log-likelihood in the M step with the IPM distance:

$$\ell(h^Y, \Phi^Y \mid p_1, \ldots, p_n) = \sum_i -p_i \log h^Y(\Phi^Y(x_i), w_i) - (1 - p_i) \log(1 - h^Y(\Phi^Y(x_i), w_i))$$
$$+ \alpha^Y \cdot \mathrm{IPM}_{\mathcal{G}^Y}\left(\left\{\Phi^Y(x_i)\right\}_{i:w_i=0}, \left\{\Phi^Y(x_i)\right\}_{i:w_i=1}\right), \quad (8)$$

where $\mathcal{G}^Y$ is a family of functions $g^Y : \mathcal{R}^Y \to \mathcal{Y}$, and $\alpha^Y$ is a hyper-parameter. For two probability density functions $p, q$ defined over $\mathcal{S} \subseteq \mathbb{R}^d$, and for a function family G of functions $g : \mathcal{S} \to \mathbb{R}$, the IPM distance is defined as $\mathrm{IPM}_{\mathrm{G}}(p, q) := \sup_{g \in \mathrm{G}} \left| \int_{\mathcal{S}} g(s)(p(s) - q(s)) ds \right|$. Similarly, we train the potential response time model using the training loss:

$$\ell(h^D, \Phi^D \mid p_1, \ldots, p_n) = \sum_{i:\tilde{y}_i^t=1} \log h^D(\Phi^D(x_i), w_i, d_i) + \sum_{i:\tilde{y}_i^t=0} p_i \log \int_{t_i}^{\infty} h^D(\Phi^D(x_i), w_i, u) du$$
$$+ \alpha^D \cdot \mathrm{IPM}_{\mathcal{G}^D}\left(\left\{\Phi^D(x_i)\right\}_{i:w_i=0}, \left\{\Phi^D(x_i)\right\}_{i:w_i=1}\right), \quad (9)$$

with $\mathcal{G}^D$ and $\alpha^D$ defined similarly. We summarize the whole algorithm including the detailed backbone and hyper-parameters choosing, as well as provide the pseudo-code in Appendix B.

# 5 EXPERIMENTS

## 5.1 BASELINES AND EVALUATION PROTOCOLS

We evaluate our framework CFR-DF, and its variant without balancing regularization (T-DF), in the task of (i) estimating HTE on the eventual outcome and (ii) estimating HTE on the response time in the always-positive stratum. We compare our method with the following methods: **T-learner** (Künzel et al., 2019), representation-based algorithms including **CFR** (Shalit et al., 2017), **SITE** (Yao et al., 2018), **Dragonnet** (Shi et al., 2019), **CFR-ISW** (Hassanpour & Greiner, 2019), **DR-CFR** (Hassanpour & Greiner, 2020) and **DER-CFR** (Wu et al., 2022), and generative algorithms **CEVAE** (Louizos et al., 2017) and **GANITE** (Yoon et al., 2018). Following the previous studies (Shalit et al., 2017; Yao et al., 2018; Wu et al., 2022), we evaluate the performance of HTE estimation using $\mathrm{PEHE} = \sqrt{\frac{1}{N} \sum_{i=1}^{N} \left((\hat{y}_i(1) - \hat{y}_i(0)) - (y_i(1) - y_i(0))\right)^2}$ and $\epsilon_{\mathrm{ATE}} = \frac{1}{N} \sum_{i=1}^{N} |(\hat{y}_i(1) - \hat{y}_i(0)) - (y_i(1) - y_i(0))|$, where $\hat{y}_i$ and $y_i$ are predicted and true outcomes.

## 5.2 DATASETS

**Synthetic Datasets.** Since the true potential outcomes are rarely available for real-world, we conduct simulation studies using synthetic datasets as follows. The observed covariates are generated from $X \sim \mathcal{N}(0, I_{m_X})$, where $I_{m_X}$ denotes $m_X$-degree identity matrix. The observed treatment $W \sim \mathrm{Bern}(\pi(X))$, where $\pi(X) = \mathbb{P}(W = 1 \mid X) = \sigma(\theta_W \cdot X)$, $\theta_W \sim U(-1, 1)$, and $\sigma(\cdot)$ denotes the sigmoid function. For the eventual potential outcomes, we generate the control outcome $Y(0) \sim \mathrm{Bern}(\sigma(\theta_{Y0} \cdot X^2 + 1))$, and the treated outcome $Y(1) \sim \mathrm{Bern}(\sigma(\theta_{Y1} \cdot X^2 + 2))$, where $\theta_{Y0}, \theta_{Y1} \sim U(-1, 1)$. In addition, we generate the potential response time $D(0) \sim \mathrm{Exp}(\exp(\theta_{D0} \cdot X)^{-1})$, and $D(1) \sim \mathrm{Exp}(\exp(\theta_{D1} \cdot X - b_D)^{-1})$, where $\theta_{D0}, \theta_{D1} \sim U(-0.1, 0.1)$, and $b_D$ *controls the heterogeneity of response time functions*. The observation time is generated via $T \sim \mathrm{Exp}(\lambda)$, where $\lambda$ is the rate parameter of the exponential distribution, and we set $\lambda = 1$ in our experiments, i.e., the average observation time is $\bar{T} = \lambda^{-1} = 1$. Finally, the observed outcome is $\tilde{Y}^T(W) = W \cdot Y(1) \cdot \mathbb{I}(T \geq D(1)) + (1 - W) \cdot Y(0) \cdot \mathbb{I}(T \geq D(0))$, where $\mathbb{I}(\cdot)$ is the indicator function.

From the data generation process described above, we sample $N = 20,000$ samples for training and $3,000$ samples for testing. We repeat each experiment 10 times to report the mean and standard

Table 2: Performance comparison (MSE $\pm$ SD) on synthetic datasets with varying $b_D$.

| Method | TOY ($b_D = 0$) PEHE | $\epsilon_{\mathrm{ATE}}$ | TOY ($b_D = 0.5$) PEHE | $\epsilon_{\mathrm{ATE}}$ | TOY ($b_D = 1$) PEHE | $\epsilon_{\mathrm{ATE}}$ |
|---|---|---|---|---|---|---|
| T-learner | $0.535 \pm 0.041$ | $0.069 \pm 0.024$ | $0.514 \pm 0.036$ | $0.028 \pm 0.017$ | $0.523 \pm 0.028$ | $0.109 \pm 0.017$ |
| CFR | $0.536 \pm 0.042$ | $0.071 \pm 0.025$ | $0.517 \pm 0.037$ | $0.025 \pm 0.016$ | $0.523 \pm 0.028$ | $0.108 \pm 0.016$ |
| SITE | $0.630 \pm 0.058$ | $0.023 \pm 0.041$ | $0.646 \pm 0.077$ | $0.026 \pm 0.020$ | $0.654 \pm 0.039$ | $0.128 \pm 0.045$ |
| Dragonnet | $0.612 \pm 0.080$ | $0.101 \pm 0.055$ | $0.499 \pm 0.023$ | $0.028 \pm 0.024$ | $0.504 \pm 0.018$ | $0.095 \pm 0.032$ |
| CFR-ISW | $0.552 \pm 0.057$ | $0.064 \pm 0.040$ | $0.602 \pm 0.084$ | $0.034 \pm 0.024$ | $0.590 \pm 0.081$ | $0.122 \pm 0.023$ |
| DR-CFR | $0.539 \pm 0.030$ | $0.071 \pm 0.032$ | $0.521 \pm 0.044$ | $0.032 \pm 0.026$ | $0.524 \pm 0.038$ | $0.107 \pm 0.035$ |
| DER-CFR | $0.548 \pm 0.051$ | $0.051 \pm 0.029$ | $0.540 \pm 0.037$ | $0.066 \pm 0.043$ | $0.568 \pm 0.034$ | $0.162 \pm 0.032$ |
| CEVAE | $0.661 \pm 0.077$ | $0.123 \pm 0.039$ | $0.661 \pm 0.077$ | $0.122 \pm 0.039$ | $0.661 \pm 0.077$ | $0.122 \pm 0.039$ |
| GANITE | $0.672 \pm 0.074$ | $0.173 \pm 0.037$ | $0.662 \pm 0.075$ | $0.147 \pm 0.036$ | $0.655 \pm 0.076$ | $0.122 \pm 0.035$ |
| T-DF | $\underline{0.416 \pm 0.019}$ | $\underline{0.021 \pm 0.008}$ | $\underline{0.432 \pm 0.013}$ | $\underline{0.017 \pm 0.014}$ | $\underline{0.407 \pm 0.016}$ | $\underline{0.013 \pm 0.007}$ |
| CFR-DF | $\mathbf{0.409 \pm 0.018}$ | $\mathbf{0.019 \pm 0.008}$ | $\mathbf{0.404 \pm 0.014}$ | $\mathbf{0.013 \pm 0.009}$ | $\mathbf{0.395 \pm 0.013}$ | $\mathbf{0.011 \pm 0.009}$ |

Table 3: PEHE of HTE estimations for potential response times with varying $b_D$.

| TOY ($b_D = 0$) | $\mathbb{P}(D(1) > d \mid Y(0) = 1, Y(1) = 1, X = x) - \mathbb{P}(D(0) > d \mid Y(0) = 1, Y(1) = 1, X = x)$ | | | | | | $\tau_D(x)$ |
|---|---|---|---|---|---|---|---|
| $D > d$ | $d = 0.1$ | $d = 0.2$ | $d = 0.5$ | $d = 1.0$ | $d = 2.0$ | $d = 5.0$ | N/A |
| T-DF | $0.017 \pm 0.003$ | $0.031 \pm 0.005$ | $0.056 \pm 0.009$ | $0.068 \pm 0.012$ | $0.055 \pm 0.012$ | $0.015 \pm 0.007$ | $0.190 \pm 0.030$ |
| CFR-DF | $\mathbf{0.014 \pm 0.001}$ | $\mathbf{0.025 \pm 0.003}$ | $\mathbf{0.045 \pm 0.005}$ | $\mathbf{0.054 \pm 0.007}$ | $\mathbf{0.042 \pm 0.005}$ | $\mathbf{0.008 \pm 0.002}$ | $\mathbf{0.152 \pm 0.016}$ |
| TOY ($b_D = 1$) | $\mathbb{P}(D(1) > d \mid Y(0) = 1, Y(1) = 1, X = x) - \mathbb{P}(D(0) > d \mid Y(0) = 1, Y(1) = 1, X = x)$ | | | | | | $\tau_D(x)$ |
| $D > d$ | $d = 0.1$ | $d = 0.2$ | $d = 0.5$ | $d = 1.0$ | $d = 2.0$ | $d = 5.0$ | N/A |
| T-DF | $0.025 \pm 0.004$ | $0.040 \pm 0.007$ | $0.055 \pm 0.010$ | $0.054 \pm 0.013$ | $0.041 \pm 0.014$ | $0.012 \pm 0.007$ | $0.321 \pm 0.056$ |
| CFR-DF | $\mathbf{0.024 \pm 0.003}$ | $\mathbf{0.037 \pm 0.005}$ | $\mathbf{0.048 \pm 0.005}$ | $\mathbf{0.043 \pm 0.006}$ | $\mathbf{0.030 \pm 0.006}$ | $\mathbf{0.006 \pm 0.002}$ | $\mathbf{0.314 \pm 0.047}$ |

deviation of the results (PEHE and $\epsilon_{\mathrm{ATE}}$). Moreover, we vary the heterogeneity of response times by setting $b_D \in \{0, 0.5, 1\}$, named the dataset as TOY ($b_D = 0$), TOY ($b_D = 0.5$), and TOY ($b_D = 1$), respectively. Besides, we evaluate our algorithm on the TOY ($b_D = 0$) and TOY ($b_D = 1$) with the average observation time $\bar{T} \in \{0.5, 1, 5, 10, 20, 50\}$.

**Real-World Datasets.** We also evaluate our CFR-DF on three widely-adopted real-world datasets: AIDS (Hammer et al., 1997; Norcliffe et al., 2023) containing 1,156 samples with 11 variables, JOBS (LaLonde, 1986; Shalit et al., 2017) containing 3,212 samples with 17 variables, and TWINS (Almond et al., 2005; Wu et al., 2022) containing 11,400 samples with 39 variables, from which we obtain covariates $X$. More details on each dataset can be found in Appendix C.1. Following the same procedure for generating synthetic datasets, we generate treatment $W$, potential outcomes $Y(0)$ and $Y(1)$, potential response times $D(0)$ and $D(1)$, observation time $T$ and factual outcomes $\tilde{Y}^T(W)$. Then, we randomly split the samples into training/testing with an 80/20 ratio with 10 repetitions.

## 5.3 RESULTS

**Performance Comparison.** We compare our method with the baselines for estimating the HTE on the eventual outcome with varying response time functions in Table 2. The optimal and second-optimal performance are **bold** and underlined, respectively. First, the proposed CFR-DF stably outperforms the baselines, as the previous methods do not take into account the delayed response, leading to biased estimates of HTE. Second, the T-DF method without using balancing regularization slightly degrades the performance compared to CFR-DF, due to the inability to resolve the confounding bias from covariate shift. Third, we observe a decrease in PEHE and ATE of 23% and 17% in TOY ($b_D = 0$), 21% and 48% in TOY ($b_D = 0.5$), and 46% and 88% in TOY ($b_D = 1$), respectively, when comparing our CFR-DF method to the optimal baseline method. These results highlight the scalability of our method to different levels of observation times, demonstrating its potential for real-world applications. Table 3 shows the performance of our methods in estimating HTE on the response times, as described in Section 3.2. We report the PEHE on estimating $\mathbb{P}(D(1) > d \mid Y(0) = 1, Y(1) = 1, X = x) - \mathbb{P}(D(0) > d \mid Y(0) = 1, Y(1) = 1, X = x)$ and $\tau_D(x)$, respectively, where the former has a more fine-grained description with varying $d$. We find that both T-DF and CFR-DF can effectively estimate the treatment effect on response time, and the CFR-DF with balancing regularization stably performs better, again demonstrating the need to adjust for confounding bias. In Appendix C.2, we further conduct the experiments with various number of features.

**Ablation Studies.** Figure 2 compares the proposed CFR-DF and its ablated versions for estimating HTE on the eventual outcome with varying average observation time, where T-DF does not perform

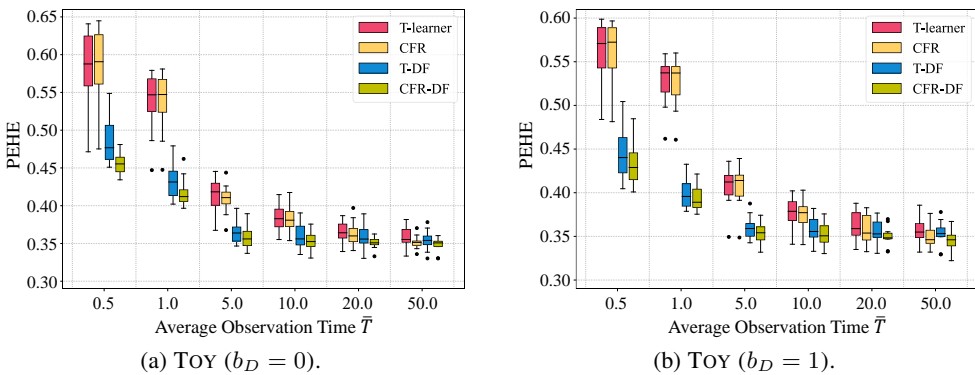

Figure 2: Effects of varying average observation time on synthetic datasets with varying $b_D$.

Table 4: Performance comparison (MSE $\pm$ SD) on JOBS and TWINS datasets.

| Method | AIDS | | JOBS | | TWINS | |
|---|---|---|---|---|---|---|
| | PEHE | $\epsilon_{ATE}$ | PEHE | $\epsilon_{ATE}$ | PEHE | $\epsilon_{ATE}$ |
| T-learner | 0.525 ± 0.052 | 0.091 ± 0.064 | 0.528 ± 0.043 | 0.085 ± 0.041 | 0.390 ± 0.071 | 0.050 ± 0.029 |
| CFR | 0.531 ± 0.046 | 0.083 ± 0.058 | 0.510 ± 0.035 | 0.064 ± 0.039 | 0.378 ± 0.057 | 0.029 ± 0.018 |
| SITE | 0.601 ± 0.031 | 0.082 ± 0.056 | 0.568 ± 0.045 | 0.064 ± 0.053 | 0.495 ± 0.087 | 0.139 ± 0.053 |
| Dragonnet | 0.546 ± 0.051 | 0.105 ± 0.042 | 0.555 ± 0.060 | 0.084 ± 0.060 | 0.440 ± 0.103 | 0.096 ± 0.067 |
| CFR-ISW | 0.592 ± 0.053 | 0.098 ± 0.032 | 0.499 ± 0.035 | 0.058 ± 0.056 | 0.392 ± 0.048 | 0.039 ± 0.023 |
| DR-CFR | 0.577 ± 0.056 | 0.078 ± 0.044 | 0.525 ± 0.077 | 0.079 ± 0.060 | 0.390 ± 0.046 | 0.039 ± 0.027 |
| DER-CFR | 0.609 ± 0.076 | 0.081 ± 0.074 | 0.503 ± 0.037 | 0.072 ± 0.043 | 0.398 ± 0.068 | 0.080 ± 0.066 |
| CEVAE | 0.623 ± 0.042 | 0.143 ± 0.019 | 0.638 ± 0.062 | 0.102 ± 0.058 | 0.526 ± 0.055 | 0.139 ± 0.027 |
| GANITE | 0.605 ± 0.034 | 0.136 ± 0.020 | 0.629 ± 0.053 | 0.151 ± 0.067 | 0.509 ± 0.056 | 0.139 ± 0.040 |
| T-DF | 0.521 ± 0.042 | 0.077 ± 0.030 | 0.453 ± 0.066 | 0.058 ± 0.030 | 0.366 ± 0.027 | 0.030 ± 0.018 |
| CFR-DF | **0.499 ± 0.055** | **0.073 ± 0.031** | **0.438 ± 0.059** | **0.051 ± 0.031** | **0.357 ± 0.017** | **0.027 ± 0.015** |

balancing regularization, CFR does not consider delayed response, and neither is considered for T-learner. We have the following findings. The proposed CFR-DF and T-DF have significantly better performance when the observation time is shorter, due to their effective adjustment for delayed response. When increasing the average observation time leads to more delayed responses being observed, we find improved performance for all four methods. The PEHE of CFR-DF stabilizes when the average observation time is above 5, and the variance gradually decreases with increasing observation time. When the observation time reaches 50, meaning all delayed responses have been observed, our method performs similarly to the CFR algorithm, and T-DF is degenerate to T-learner.

**Real-World Experiments.** We conduct real-world experiments using AIDS, JOBS and TWINS datasets. The AIDS (Hammer et al., 1997) contains people with HIV and SEER with Prostate Cancer. The JOBS dataset (LaLonde, 1986) is based on the National Supported Work program and examines the effects of job training on income and employment status after training. The TWINS dataset (Almond et al., 2005) studies the effects of infant weight on the death rate. Notably, job training takes time to cause changes in incomes, and infants also take time to observe their mortality outcomes (and thus study the effect on mortality), therefore it is reasonable to study the delayed response in such real-world applications. Table 4 demonstrates that the proposed CFR-DF algorithm outperforms all baselines on these real-world datasets, showcasing its effectiveness.

## 6 CONCLUSION

This paper studies the HTE estimation problem by further considering the response time needed for a treatment to produce a causal effect on the outcome. Specifically, we propose a principled learning algorithm, called CFR-DF, to estimate both eventual potential outcomes and potential response times. Considering the widespread of delayed feedback outcomes, we believe such study is meaningful for real-world applications. A shortcoming of our study is the validity of the assumptions in practice, e.g., we need enough observation time to identify HTE on the eventual potential outcome, and principal ignorability is further required to identify HTE on the response time. Studying how to weaken these assumptions, and identifying and estimating HTE with delayed responses are served as future topics.

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

## A  Theorems and Proofs

### A.1  The Proofs of Theorems 1 and 2

First, we recap the assumptions in Section 3.3 as below. Next, we provide formal proofs of Theorem 1, Lemma 1, and Theorem 2, respectively.

**Assumption 1** (Unconfoundedness). $W \perp\!\!\!\perp (D(0), D(1), \tilde{Y}^t(0), \tilde{Y}^t(1)) \mid X$ *for all* $t > 0$.

**Assumption 2** (Time Independence). $T \perp\!\!\!\perp (D(0), D(1), \tilde{Y}^t(0), \tilde{Y}^t(1), W) \mid X$ *for all* $t > 0$.

**Assumption 3** (Time Sufficiency). $\inf\{d : F_D^{(w)}(d \mid Y(w) = 1, X) = 1\} < \inf\{t : F_T(t) = 1\}$ *for* $w = 0, 1$, *where* $F(\cdot)$ *is the cumulative distribution function (cdf).*

**Assumption 4** (Monotonicity). $Y(0) \leq Y(1)$.

**Assumption 5** (Principal Ignorability). $(W, Y(w)) \perp\!\!\!\perp D(1-w) \mid Y(1-w), X$ *for* $w = 0, 1$.

**Theorem 1.** *Under Assumptions 1-3, the HTE on the eventual outcome* $\tau(x)$ *is identifiable.*

*Proof of Theorem 1.* For units with $Y(w) = 0$, we set $D(w) = \infty$, for $w = 0, 1$. We first prove the identifiability of $\mathbb{P}(D(w) > t \mid X = x)$ for $w = 0, 1$ and $t > 0$. Under Assumption 1, we have:

$$
\begin{aligned}
-\frac{d}{dt} \log \mathbb{P}(D(w) > t \mid X = x) &= \lim_{h \to 0^+} \frac{\frac{1}{h}\mathbb{P}(t < D(w) \leq t + h \mid X = x)}{P(D(w) > t \mid X = x)} \\
&= \lim_{h \to 0^+} \frac{\frac{1}{h}\mathbb{P}(t < D(w) \leq t + h \mid W = w, X = x)}{\mathbb{P}(D(w) > t \mid W = w, X = x)} \\
&= \lim_{h \to 0^+} \frac{1}{h}\mathbb{P}(t < D(w) \leq t + h \mid W = w, X = x, D(w) > t),
\end{aligned}
\tag{10}
$$

where the first equality follows from the definition of first-order derivative, the second equality follows from the unconfoundedness assumption, and the third equality follows from the definition of conditional probability. Under Assumption 2, we obtain the identifiability result in the following:

$$
\begin{aligned}
&\lim_{h \to 0^+} \frac{1}{h}\mathbb{P}(t < D(w) \leq t + h \mid W = w, X = x, D(w) > t) \\
=&\lim_{h \to 0^+} \frac{1}{h}\mathbb{P}(t < D(w) \leq t + h \mid W = w, X = x, D(w) > t, T > t) \\
=&\lim_{h \to 0^+} \frac{1}{h}\mathbb{P}(t < \min\{D(w), T\} \leq t + h, \mathbb{I}(D(w) \leq T) = 1 \mid W = w, X = x, \min\{D(w), T\} > t) \\
=&\lim_{h \to 0^+} \frac{1}{h}\mathbb{P}(t < \min\{D, T\} \leq t + h, \mathbb{I}(D \leq T) = 1 \mid W = w, X = x, \min\{D, T\} > t),
\end{aligned}
\tag{11}
$$

where the first equality follows from the time independence assumption, the second equality follows from the equivalence between $t < D(w) \leq t + h$ and $t < \min\{D(w), T\} \leq t + h$ and $D(w) \leq T$, given the condition that $T > t$ with a sufficiently small time period $h \to 0^+$, the third equality follows from the unconfoundedness assumption. Also, we can identify:

$$
\mathbb{P}(D(w) > t \mid X = x) = \exp\left\{\int_0^t \frac{d}{du} \log \mathbb{P}(D(w) > u \mid X = x) du\right\}
\tag{12}
$$

for $w = 0, 1$, because we have prove the identifiability of $-\frac{d}{dt} \log \mathbb{P}(D(w) > t \mid X = x)$.

We next show the identifiability of $\mathbb{P}(Y(w) = 1 \mid X = x)$. Under Assumption 3, we have

$$
\begin{aligned}
\mathbb{P}(Y(w) = 1 \mid X = x) &= 1 - \mathbb{P}(Y(w) = 0 \mid X = x) = 1 - \lim_{t \to \infty} \mathbb{P}(D(w) > t \mid X = x) \\
&= 1 - \mathbb{P}(D(w) > q_d \mid X = x) = 1 - \mathbb{P}(D(w) > q \mid X = x)
\end{aligned}
\tag{13}
$$

for $q_d \leq q < q_t$, where $q_d = \inf\left\{d : F_D^{(w)}(d \mid Y(w) = 1, X) = 1\right\}$, $q_t = \inf\{t : F_T(t) = 1\}$ and $F(\cdot)$ is the cumulative distribution function (cdf). Therefore, $\mathbb{P}(Y(w) = 1 \mid X = x)$ is identifiable from observed data for $w = 0, 1$. $\square$

**Lemma 1.** *Under Assumptions 1-4, $\mathbb{P}(Y(0) = 1, Y(1) = 1 \mid X = x)$ is identifiable.*

*Proof of Lemma 1.* Under Assumption 4, we have

$$
\begin{aligned}
\mathbb{P}(Y(0) = 0, Y(1) = 0 \mid X = x) &= \mathbb{P}(Y(1) = 0 \mid X = x) \\
\mathbb{P}(Y(0) = 0, Y(1) = 1 \mid X = x) &= \mathbb{P}(Y(1) = 1 \mid X = x) - \mathbb{P}(Y(0) = 1 \mid X = x) \\
\mathbb{P}(Y(0) = 1, Y(1) = 1 \mid X = x) &= \mathbb{P}(Y(0) = 1 \mid X = x).
\end{aligned}
\tag{14}
$$

Then the identifiability of the left-hand side parameters follows directly from the identifiability of $\mathbb{P}(Y(w) = 1 \mid X = x)$ for $w = 0, 1$ under Assumptions 1-3 as shown in Theorem 1. $\qquad \square$

**Theorem 2.** *Under Assumptions 1-5, the HTE on the response time in the always-positive stratum $\tau_D(x) = \mathbb{E}[D(1) - D(0) \mid Y(0) = 1, Y(1) = 1, X = x]$ is identifiable.*

*Proof of Theorem 2.* Under Assumption 5, i.e., $(W, Y(0)) \perp\!\!\!\perp D(1) \mid Y(1), X$, we have

$$
\begin{aligned}
&\mathbb{P}(D(1) < t \mid Y(0) = 1, Y(1) = 1, X = x) = \mathbb{P}(D(1) < t \mid Y(1) = 1, X = x) \\
&= \mathbb{P}(D(1) < t \mid Y(1) = 1, X = x, W = 1) = \mathbb{P}(D(1) < t \mid Y = 1, X = x, W = 1) \\
&= \frac{\mathbb{P}(D < t \mid X = x, W = 1)}{\mathbb{P}(Y = 1 \mid X = x, W = 1)} \\
&= \frac{1 - \exp\left\{\int_0^t \frac{d}{du} \log \mathbb{P}(D(1) > u \mid X = x) du\right\}}{1 - \lim_{t \to \infty} \exp\left\{\int_0^t \frac{d}{du} \log \mathbb{P}(D(1) > u \mid X = x) du\right\}},
\end{aligned}
\tag{15}
$$

which is identifiable, because we have proved the identifiability of $-\frac{d}{dt} \log \mathbb{P}(D(1) > t \mid X = x)$ in Theorem 1. Similarly, we can identify

$$
\mathbb{P}(D(0) < t \mid Y(0) = 1, Y(1) = 1, X = x) = \frac{1 - \exp\left\{\int_0^t \frac{d}{du} \log \mathbb{P}(D(0) > u \mid X = x) du\right\}}{1 - \lim_{t \to \infty} \exp\left\{\int_0^t \frac{d}{du} \log \mathbb{P}(D(0) > u \mid X = x) du\right\}}.
\tag{16}
$$

Then $\tau_D(x)$ is identifiable due to

$$
\begin{aligned}
\tau_D(x) &= \mathbb{E}[D(1) - D(0) \mid Y(0) = 1, Y(1) = 1, X = x] \\
&= -\int_0^\infty \mathbb{P}(D(1) < u \mid Y(0) = 1, Y(1) = 1, X = x) du \\
&\quad + \int_0^\infty \mathbb{P}(D(0) < u \mid Y(0) = 1, Y(1) = 1, X = x) du.
\end{aligned}
\tag{17}
$$

$\qquad \square$

## A.2 THE COMPUTATION DETAILS OF PARAMETRIC AND NON-PARAMETRIC POTENTIAL RESPONSE TIME MODELS

In this paper, we propose a principled learning approach called CFR-DF (**C**ounter**F**actual **R**egression with **D**elayed **F**eedback) that simultaneously predicts potential outcomes and potential response times by employing an EM algorithm with eventual outcomes treated as latent variables. Due to space limitations, we only provide the explicit solutions of the EM algorithm in a general functional form for estimating the parameters of interest in Section 4 in the main text. However, in practice, empirical computation requires model specification: either (i) a parametric model or (ii) a non-parametric model based on weighted kernel functions.

**Parametric model**: One can assume that the potential delayed response times obey exponential models for both treatment and control groups. Specifically, let $\mathbb{P}(D(w) = u \mid X = \mathbf{x}, Y(w) = 1) = \lambda_w(\mathbf{x}) \exp(-\lambda_w(\mathbf{x})u)$ for $w = 0, 1$. Then we have: $\int_t^\infty \mathbb{P}(D(w) = u \mid X = \mathbf{x}, Y(w) = 1) du = \int_t^\infty \lambda_w(\mathbf{x}) \exp(-\lambda_w(\mathbf{x})u) du = \exp(-\lambda_w(\mathbf{x})t)$ in the derived $p_i$ in the E-step.

**Non-parametric model based on weighted kernel functions**: potential delayed response times can be further extended to a nonparametric model using a set of weighted kernel functions. Specifically,

let the non-parametric hazard function is $h_w(d; \mathbf{x}) = \sum_{l=1}^{L} \alpha_l^w(\mathbf{x}) k(t_l, d)$ for $w = 0, 1$, where $k$ is a kernel function returning a positive value, and intuitively represents the similarity between two time points. Here, one can use kernel functions as $k$ such that $k(t_l, u)$, $\int_0^a k(t_l, u)\, du$ and $\int_a^\infty k(t_l, u)\, du$ for $t_l, u, a \geq 0$ can be calculated analytically.

For example, a Gaussian kernel with bandwidth parameter $h > 0$ leads to $k(t_l, u) = \exp\left(-\frac{(t_l - u)^2}{2h^2}\right)$, $\int_0^a k(t_l, u)\, du = -h\sqrt{\frac{\pi}{2}} \left[\mathrm{erf}\left(\frac{t_l - a}{\sqrt{2}h}\right) - \mathrm{erf}\left(\frac{t_l}{\sqrt{2}h}\right)\right]$, and $\int_a^\infty k(t_l, u)\, du = h\sqrt{\frac{\pi}{2}} \left[1 + \mathrm{erf}\left(\frac{t_l - a}{\sqrt{2}h}\right)\right]$, where leads to the analytical form $p_i$ in the E-step.

Given the hidden variable values $p_i$ computed from the E-step, we can plug them into the expected log-likelihood during the M-step:

$$
\sum_{i:\tilde{y}_i^t = 1} \log \mathbb{P}(\tilde{Y}_i^T = 1, D = d_i \mid X = x_i, W = w_i, T = t_i)
$$

$$
+ \sum_{i:\tilde{y}_i^t = 0} (1 - p_i) \log \mathbb{P}(\tilde{Y}_i^T = 0, Y_i(w_i) = 0 \mid X = x_i, W = w_i, T = t_i)
$$

$$
+ \sum_{i:\tilde{y}_i^t = 0} p_i \log \mathbb{P}(\tilde{Y}_i^T = 0, Y_i(w_i) = 1 \mid X = x_i, W = w_i, T = t_i). \tag{18}
$$

From a similar argument as derived above, the expected log-likelihood is equal to:

$$
\sum_i p_i \log \mathbb{P}(Y_i(w_i) = 1 \mid X = x_i) + (1 - p_i) \log(1 - \mathbb{P}(Y_i(w_i) = 1 \mid X = x_i))
$$

$$
+ \sum_{i:\tilde{y}_i^t = 1} \log \mathbb{P}(D_i(w_i) = d_i \mid X = x_i, Y_i(w_i) = 1)
$$

$$
+ \sum_{i:\tilde{y}_i^t = 0} p_i \log \int_{t_i}^\infty \mathbb{P}(D(w_i) = u \mid X = x_i, Y_i(w_i) = 1)\, du, \tag{19}
$$

in which the eventual potential outcome model $\mathbb{P}(Y(w) = 1 \mid X = x)$ and the potential response time model $\mathbb{P}(D(w) = d \mid X = x, Y(w) = 1)$ can be optimized independently. Notably, in our experiments, we used *Parametric models* for delay time modeling in the treated and control groups.

## B  ALGORITHM, HYPER-PARAMETERS AND DISCUSSION

### B.1  ALGORITHM DETAILS AND ENVIRONMENT CONFIGURATION

**Motivation**: In this paper, we study the problem of estimating HTE with a delayed response, which can be seen as a censoring problem with imbalanced treatment assignment: the observation time $T$ refers to the "time-to-censor", the response time $D$ refers to the "time-to-event", and the treatment is not assigned at random. We must emphasize that simply applying the expectation-maximization technique is insufficient to recover the delayed outcome without making additional assumptions and identification guarantees. Because this problem involves not only missing data but also survival analysis and confounding bias. To address these issues, we propose a novel CFR-DF approach that extends counterfactual regression to delayed feedback outcomes using a modified EM algorithm with identification guarantees. In Appendix A.2, we provide the explicit solutions of the EM algorithm with model specification: either (i) a parametric model or (ii) a non-parametric model based on weighted kernel functions. In our experiments, we use *Parametric models* for delay time modeling in the treated and control groups. Algorithm 1 shows the pseudo-code of our CFR-DF.

**Architecture**: In the CFR-DF architecture (Figure 3), we use three-layer neural networks $\Phi_0^Y$ and $\Phi_1^Y$ with ELU activation function and BatchNorm to learn representation of the eventual outcome, and two-layer neural networks $\Phi_0^D$ and $\Phi_1^D$ with ELU activation function and BatchNorm to learn representation of the delayed response time. Each layer in these networks consists of $m_X$ neural units. Then, we use a single-layer network $h^Y$ with Sigmoid activation to achieve $\hat{P}(Y = 1)$ and a single-layer network $h^D$ with SoftPlus sigmoid activation to achieve $\hat{\lambda}$. Dropout is not utilized in the

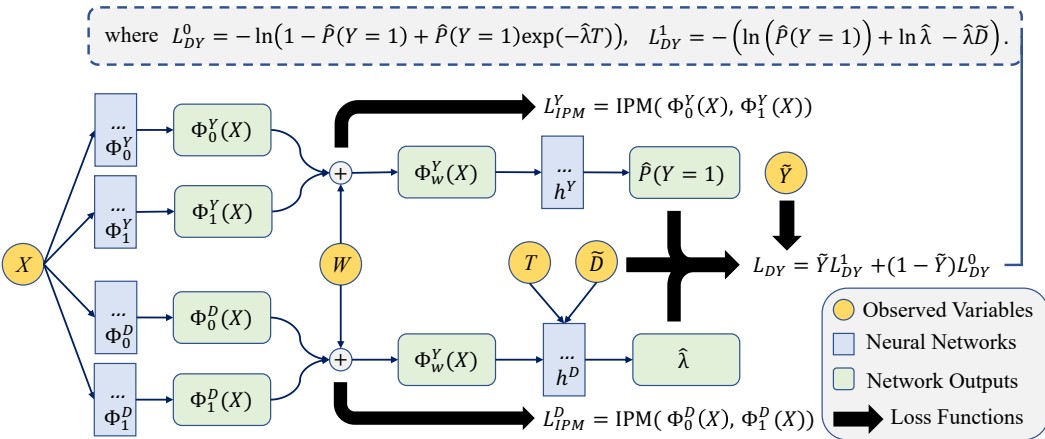

Figure 3: Overview of CFR-DF Architecture. For the representation block, we use multi-layer neural networks $\Phi$ with ELU activation function to learn representation and each network has two/three layers with $m_X$ units, respectively. Then, we use a single-layer network $h^Y$ with Sigmoid activation to achieve $\hat{P}(Y = 1)$ and a single-layer network $h^D$ with SoftPlus sigmoid activation to achieve $\hat{\lambda}$.

---

**Algorithm 1** CounterFactual Regression with Delayed Feedback Outcomes (CFR-DF)

---

**Input:** Observational data $\mathbb{D} = \{x_i, w_i, t_i, \tilde{d}_i, \tilde{y}_i\}_{i=1}^n$ (we set $\tilde{d}_i = -1$ for all subjects with $\tilde{y}_i = 0$ in training process); hyper-parameters $\alpha^Y$ and $\alpha^D$; neural networks $\{\Phi_0^Y(\cdot), \Phi_1^Y(\cdot), \Phi_0^D(\cdot), \Phi_1^D(\cdot), h^D(\cdot), h^Y(\cdot)\}$; maximum number of iterations $M = 3000$; stopping criterion $\epsilon = 0.002$; initiation loss $L_{s=0} = 9999.9$; and iteration counter $s = 0$.
**Output:** $\hat{P}_i(Y = 1) = h^Y(\Phi^Y(x_i), w_i), \quad \hat{d}_i = \hat{\lambda}_i^{-1}, \quad \hat{\lambda}_i = h^D(\Phi^D(x_i), w_i, t_i)$.
**Loss function:** $L = \tilde{Y} \cdot L_{DY}^1 + (1 - \tilde{Y}) \cdot L_{DY}^0 + \alpha^D \cdot L_{IPM}^D + \alpha^Y \cdot L_{IPM}^Y$.
**CFR-DF:**
$s \leftarrow s + 1$;
$L_s = \tilde{Y} \cdot L_{DY}^1 + (1 - \tilde{Y}) \cdot L_{DY}^0 + \alpha^D \cdot L_{IPM}^D + \alpha^Y \cdot L_{IPM}^Y$;
**while** $s \le M$ and $|L_s - L_{s-1}| > \epsilon$ **do**
  $s \leftarrow s + 1$;
  $\Phi^Y(x_i) = w_i \Phi_1^Y(x_i) + (1 - w_i)\Phi_0^Y(x_i), \quad \Phi^D(x_i) = w_i \Phi_1^D(x_i) + (1 - w_i)\Phi_0^D(x_i)$;
  $\hat{P}_i(Y = 1) = h^Y(\Phi^Y(x_i), w_i), \quad \hat{\lambda}_i = h^D(\Phi^D(x_i), w_i, t_i)$;
  $L_{IPM}^Y = \text{IPM}\left(\{\Phi^Y(x_i)\}_{i:w_i=0}, \{\Phi^Y(x_i)\}_{i:w_i=1}\right)$;
  $L_{IPM}^D = \text{IPM}\left(\{\Phi^D(x_i)\}_{i:w_i=0}, \{\Phi^D(x_i)\}_{i:w_i=1}\right)$;
  $L_{DY}^0(x_i) = -\ln(1 - \hat{P}_i(Y = 1) + \hat{P}_i(Y = 1)\exp(-\hat{\lambda}_i t_i))$;
  $L_{DY}^1(x_i) = -(\ln(\hat{P}_i(Y = 1)) + \ln \hat{\lambda}_i - \hat{\lambda}_i \tilde{d}_i)$;
  $L_s = \frac{1}{n} \cdot \sum_{i=1}^n \left(\hat{y}_i L_{DY}^1(x_i) + (1 - \hat{y}_i) L_{DY}^0(x_i)\right) + \alpha^D \cdot L_{IPM}^D + \alpha^Y \cdot L_{IPM}^Y$;
  Update $\{\Phi_0^Y, \Phi_1^Y, \Phi_0^D, \Phi_1^D, h^D, h^Y\} \leftarrow \text{Adam}\{L_s\}$;
**end while**

---

CFR-DF architecture, but BatchNorm is applied in each layer of the representation networks. Finally, we update $\{\Phi_0^Y, \Phi_1^Y, \Phi_0^D, \Phi_1^D, h^D, h^Y\}$ using Adam $L_s$ optimizer.

**Hardware used**: Ubuntu 16.04.3 LTS operating system with 2 * Intel Xeon E5-2660 v3 @ 2.60GHz CPU (40 CPU cores, 10 cores per physical CPU, 2 threads per core), 256 GB of RAM, and 4 * GeForce GTX TITAN X GPU with 12GB of VRAM.

**Software used**: Python 3.8 with numpy 1.24.2, pandas 2.0.0, pytorch 2.0.0.

### B.2 HYPER-PARAMETER OPTIMIZATION

In this paper, we adopt an early stopping criterion ($\varepsilon$) to select the best-evaluated iterate for each model. The hyper-parameters $\alpha^Y$ and $\alpha^D$ are selected from a range of values $\{1e - 4, 5e - 4, 1e -$

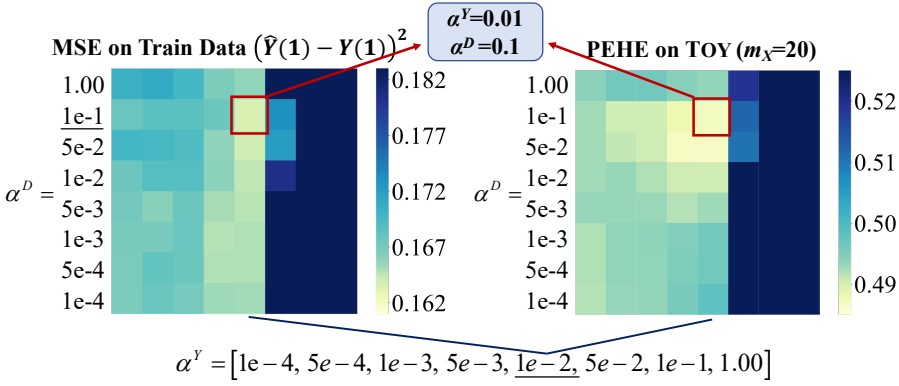

Figure 4: Hyper-Parameter Optimization: The smallest MSE on Train Data implies the best Hyper-Parameters. The optimal hyper-parameters are $\alpha^Y = 0.01, \alpha^{D=0.1}$ for $\text{TOY}(m_X = 20)$.

$3, 5e - 3, 1e - 2, 5e - 2, 1e - 1, 1.00\}$ based on the mean squared error (MSE) of $Y(1)$ on the training data. We optimize the hyper-parameters in CFR-DF by minimizing the objective loss on the training data. Taking $\text{TOY}(m_X = 20)$ as an example, as depicted in Figure 4, we determine the hyper-parameters that correspond to the smallest MSE $(\widehat{Y}(1) - Y(1))^2$ on the training data, which indicates the optimal hyper-parameters for PEHE on $\text{TOY}(m_X = 20)$. The optimal hyper-parameters for each dataset can be found in Table 5 in Appendix B.2.

Table 5: Optimal Hyper-Parameters.

|  | $\alpha^Y$ | $\alpha^D$ |
|---|---|---|
| $\text{TOY}(m_X = 5)$ | 0.005 | 0.1 |
| $\text{TOY}(m_X = 10)$ | 0.01 | 0.1 |
| $\text{TOY}(m_X = 20)$ | 0.01 | 0.1 |
| $\text{TOY}(m_X = 40)$ | 0.01 | 0.1 |
| AIDS | 0.01 | 0.01 |
| JOBS | 0.005 | 0.05 |
| TWINS | 0.005 | 0.01 |

Table 6: Datasets Used for Evaluation.

|  | No. instances | No. features |
|---|---|---|
| $\text{TOY}(m_X = 5)$ | 20000 | 5 |
| $\text{TOY}(m_X = 10)$ | 20000 | 10 |
| $\text{TOY}(m_X = 20)$ | 20000 | 20 |
| $\text{TOY}(m_X = 40)$ | 20000 | 40 |
| AIDS | 1156 | 11 |
| JOBS | 3212 | 17 |
| TWINS | 11400 | 39 |

### B.3 DISCUSSION ON THE SCALABILITY TO ARBITRARY FORMS OF TREATMENTS

It should be noted that our work can be naturally extended to arbitrary forms of treatments and has rigorous theoretical guarantees regarding the identifiability of true HTE in all strata, i.e., $\mathbb{E}[Y(w) \mid X = x]$ for all $w \in \mathcal{W}$. This way, by defining delayed response time $D(w)$ for all $w \in \mathcal{W}$ similarly and following a similar argument of our identifiability proof, and substitute $Y(0)$ and $Y(1)$ to $Y(w)$ for all $w \in \mathcal{W}$, the true HTE $\mathbb{E}[Y(w) \mid X = x]$ for all $w \in \mathcal{W}$ can be identified similarly. Moreover, in the proposed time-to-event based HTE problem setup with delayed responses, the outcome of interest has to be binary to ensure well-definiteness. Specifically, an event may either occur or not occur under any form of intervention (see the discussion in the previous paragraph), i.e., $Y(w) = 1$ or not $Y(w) = 0$. It is worth noting that only the former, i.e., $Y(w) = 1$, may be subject to delayed response, leading to the "false negative" samples. For the latter, $Y(w) = 0$, it is difficult to define a delayed response because this event never occurs (hence we let $D(w) = \infty$ for $Y(w) = 0$), and we will never observe "false positive" samples. To the best of our knowledge, this is the first work in the field of causal inference to consider the potential delayed response time $D(w)$ from intervention to outcome, and we theoretically prove the identifiability of true HTE in all strata. Considering the time it takes for an intervention to have an effect on an outcome, we believe this provides reasonable motivation in the causal inference community.

Table 7: Performance comparison (MSE $\pm$ SD) on synthetic datasets with varying $m_X$.

| Method | TOY ($m_X = 5$) PEHE | $\epsilon_{\text{ATE}}$ | TOY ($m_X = 10$) PEHE | $\epsilon_{\text{ATE}}$ |
|---|---|---|---|---|
| T-learner | 0.442 ± 0.028 | 0.028 ± 0.014 | 0.514 ± 0.036 | 0.028 ± 0.017 |
| CFR | 0.441 ± 0.029 | 0.029 ± 0.015 | 0.517 ± 0.037 | 0.025 ± 0.016 |
| SITE | 0.568 ± 0.039 | 0.029 ± 0.025 | 0.646 ± 0.077 | 0.026 ± 0.020 |
| Dragonnet | 0.457 ± 0.031 | 0.053 ± 0.037 | 0.499 ± 0.023 | 0.028 ± 0.024 |
| CFR-ISW | 0.463 ± 0.053 | 0.030 ± 0.022 | 0.602 ± 0.084 | 0.034 ± 0.024 |
| DR-CFR | 0.445 ± 0.033 | 0.040 ± 0.018 | 0.521 ± 0.044 | 0.032 ± 0.026 |
| DER-CFR | 0.462 ± 0.029 | 0.037 ± 0.020 | 0.540 ± 0.037 | 0.066 ± 0.043 |
| CEVAE | 0.590 ± 0.038 | 0.126 ± 0.028 | 0.661 ± 0.077 | 0.122 ± 0.039 |
| GANITE | 0.591 ± 0.036 | 0.149 ± 0.026 | 0.662 ± 0.075 | 0.147 ± 0.036 |
| T-DF | 0.353 ± 0.057 | 0.022 ± 0.023 | 0.432 ± 0.013 | 0.017 ± 0.014 |
| CFR-DF | **0.329 ± 0.022** | **0.015 ± 0.013** | **0.404 ± 0.014** | **0.013 ± 0.009** |
| | TOY ($m_X = 20$) | | TOY ($m_X = 40$) | |
| T-learner | 0.593 ± 0.015 | 0.035 ± 0.014 | 0.677 ± 0.014 | 0.041 ± 0.010 |
| CFR | 0.588 ± 0.015 | 0.036 ± 0.017 | 0.678 ± 0.014 | 0.043 ± 0.011 |
| SITE | 0.716 ± 0.030 | 0.030 ± 0.017 | 0.760 ± 0.017 | 0.041 ± 0.014 |
| Dragone | 0.596 ± 0.016 | 0.034 ± 0.009 | 0.739 ± 0.021 | 0.041 ± 0.021 |
| CFR-ISW | 0.687 ± 0.033 | 0.056 ± 0.024 | 0.763 ± 0.030 | 0.070 ± 0.031 |
| DR-CFR | 0.633 ± 0.032 | 0.047 ± 0.035 | 0.754 ± 0.028 | 0.043 ± 0.022 |
| DER-CFR | 0.665 ± 0.030 | 0.086 ± 0.032 | 0.754 ± 0.025 | 0.053 ± 0.043 |
| CEVAE | 0.722 ± 0.030 | 0.098 ± 0.016 | 0.762 ± 0.028 | 0.078 ± 0.014 |
| GANITE | 0.717 ± 0.029 | 0.081 ± 0.016 | 0.762 ± 0.027 | 0.066 ± 0.015 |
| T-DF | 0.529 ± 0.011 | 0.018 ± 0.013 | 0.633 ± 0.008 | 0.018 ± 0.011 |
| CFR-DF | **0.498 ± 0.021** | **0.017 ± 0.010** | **0.612 ± 0.007** | **0.012 ± 0.007** |

## C DATASETS AND EXPERIMENTS

### C.1 DATASETS USED FOR EVALUATION

**Synthetic Datasets.** Following the data generation process in Section 5.2, we generated data as follows. The observed covariates are generated from $X \sim \mathcal{N}(0, I_{m_X})$, where $I_{m_X}$ denotes $m_X$-degree identity matrix. The observed treatment $W \sim \text{Bern}(\pi(X))$, where $\pi(X) = \mathbb{P}(W = 1 \mid X) = \sigma(\theta_W \cdot X)$, $\theta_W \sim U(-1, 1)$, and $\sigma(\cdot)$ denotes the sigmoid function. For the eventual potential outcomes, we generate the control outcome $Y(0) \sim \text{Bern}(\sigma(\theta_{Y0} \cdot X^2 + 1))$, and the treated outcome $Y(1) \sim \text{Bern}(\sigma(\theta_{Y1} \cdot X^2 + 2))$, where $\theta_{Y0}, \theta_{Y1} \sim U(-1, 1)$. In addition, we generate the potential response time $D(0) \sim \text{Exp}(\exp(\theta_{D0} \cdot X)^{-1})$, and $D(1) \sim \text{Exp}(\exp(\theta_{D1} \cdot X - b_D)^{-1})$, where $\theta_{D0}, \theta_{D1} \sim U(-0.1, 0.1)$, where $b_D = 0.5$ *controls the heterogeneity of response time functions*. The observation time is generated via $T \sim \text{Exp}(\lambda)$, where $\lambda$ refers to the rate parameter of the exponential distribution. We set the rate parameter as $\lambda = 1$, i.e., the average observation time is $\bar{T} = \lambda^{-1} = 1$. Finally, the observed outcome is given as $\tilde{Y}^T(W) = W \cdot Y(1) \cdot \mathbb{I}(T \geq D(1)) + (1 - W) \cdot Y(0) \cdot \mathbb{I}(T \geq D(0))$, where $\mathbb{I}(\cdot)$ is the indicator function. From the data generation process described above, we sample $N = 20,000$ samples for training and $3,000$ samples for testing. We repeat each experiment 10 times to report the mean and standard deviation of the errors.

**Real-World Datasets.** In this paper, we use three wide-applied three widely-adopted real-world datasets: AIDS[1] (Hammer et al., 1997; Norcliffe et al., 2023), JOBS[2] (LaLonde, 1986; Shalit et al., 2017), and TWINS[3] (Almond et al., 2005; Wu et al., 2022). In Table 6 we provide details about the datasets used in our evaluation. The AIDS data collected between January 1996 and January 1997 involved 1,156 patients in 33 AIDS clinical trial units and 7 National Hemophilia Foundation sites in the United States and Puerto Rico, and was used to study the impact and effectiveness of antiretroviral therapy on HIV-positive patients. The JOBS benchmark is widely used in the field of causal inference.

---

[1]AIDS is available at: https://scikit-survival.readthedocs.io/

[2]JOBS is available at: http://www.fredjo.com/

[3]TWINS is available at: http://www.nber.org/data/

It is built upon randomized controlled trials and aims to assess the effects of job training programs on employment status. The TWINS is derived from all twins born in the USA between the years 1989 and 1991, and is utilized to assess the influence of birth weight on mortality within one year.

Covariates $X$ are obtained from AIDS, JOBS, and TWINS. Following the same procedure for generating synthetic datasets, we generate treatment $W$, potential outcomes $Y(0)$ and $Y(1)$, potential response times $D(0)$ and $D(1)$, observation time $T$ and factual outcomes $\tilde{Y}^T(W)$. Then we randomly split the samples into training/testing with an 80/20 ratio with 10 repetitions.

## C.2 MORE EXPERIMENTS ON VARYING FEATURE DIMENSIONS

To evaluate our CFR-DF on a wide range of scenarios, given $b_D = 0.5$, we further tune the number of features by setting $m_X \in \{5, 10, 20, 40\}$, named the dataset as TOY ($m_X = 5$), TOY ($m_X = 10$), TOY ($m_X = 20$), and TOY ($m_X = 40$), respectively.

**Performance Comparison.** Table 7 presents a comprehensive performance comparison between our proposed method and the baselines in estimating the Heterogeneous Treatment Effect (HTE) on the eventual outcome, considering varying feature dimensions. The optimal and second-optimal performances are indicated as **bold** and underlined, respectively. Consistent with the observations from Table 2, our CFR-DF consistently outperforms the baselines, demonstrating its efficacy in addressing the label noise arising from delayed responses. In contrast, previous methods that do not consider delayed responses often yield biased estimates of HTE. Additionally, the T-DF method without using balancing regularization slightly degrades the performance compared to CFR-DF, due to the inability to resolve the confounding bias from covariate shift. Overall, our method achieves significant reductions in the PEHE and $\epsilon_{ATE}$. Specifically, comparing CFR-DF to the optimal traditional causal method in the PEHE and $\epsilon_{ATE}$, we observe reductions of 25% and 46% in TOY ($m_X = 5$), 21% and 48% in TOY ($m_X = 10$), 15% and 43% in TOY ($m_X = 20$), and 10% and 70% in TOY ($m_X = 40$), respectively. These results highlight the superior performance of CFR-DF compared to the baselines and its scalability to different feature dimensions, further emphasizing its potential for accurate and robust estimation of HTE in various practical settings.

