# OpenReview forum: "Estimating Heterogeneous Treatment Effect with Delayed Response"
_ICLR.cc/2024/Conference — Submitted to ICLR 2024_

### Official Review · Reviewer_WuYz · 2023-10-17

**Soundness:** 2 fair
**Presentation:** 3 good
**Contribution:** 1 poor
**Rating:** 3
**Confidence:** 4

**Summary:**

This paper formalizes the problem of estimating treatment effects with delayed responses. While most of the literature focuses on estimating treatment effects that occur immediately after treatment administration, the authors argue that it is more reasonable to assume that treatment effects occur after an unknown time delay. The authors demonstrate identifiability for delayed treatment effect estimates and delayed treatment effect times under extended assumptions. In addition, the authors propose an extension of counterfactual regression (CFR) to account for delayed treatment effects. To this end, they use an expectation maximization algorithm to combine estimates of delayed response time with the CFR and call this algorithm the delayed feedback counterfactual regression (CFR-DF). Finally, the authors validate their method on several synthetic and semisynthetic data sets.

**Strengths:**

- The submission is a technically sound complete piece of work. The paper is clearly written and well organized.

**Weaknesses:**

- The method is fairly straightforward (and, to a large extent, similar to existing works).
- The motivation of this work is unclear: Medical studies typically consider a sufficiently large time window for the treatment response to be observed, such that in all relevant datasets *the eventual outcome is known*. Hence, there is no need for a method that considers delayed response time, as this is simply a data preprocessing issue.
- Therefore, it would be a much fairer comparison to train the baselines on the eventual outcomes. I doubt that CFR-DF outperforms baselines that are trained on correctly formatted data.
- This work is, as pointed out in the RW section, related to censored HTE estimation under time-to-event data. The authors claim that the difference of their work is that they consider observable outcomes $Y^t \in \{0,1\}$ that eventually have a positive outcome ("conversion in uplift modeling"), while the cited literature focuses on negative outcomes (such as death of a patient). However, I wonder how this makes any difference from a technical perspective, as this is purely interpretation of the user. Methodologically, this should not make any difference (e.g., whether $Y^t=1$ denotes death of a patient or a positive treatment outcome).
- The assumptions are fairly strong. Especially, Monotonicity and Principal Ignorability are quite restrictive for identifiability. Further, time independence seems unrealistic to me. Here, the authors claim that observation times may not affect potential response times, which seems plausible. However, the effect could be the other way around. In the dynamic treatment literature, it is a known issue that observation time are, in reality, intertwined with the patient's health condition (known as "informative sampling").
- There are methods available for long term outcome targeting where short term proxies are available (which is, admittedly, a slightly different setting). IMHO it would be fair to acknowledge this literature stream. I believe this scenario is more realistic in practice.
- A few minor concerns:
	a. The authors use an exponential distribution to model the response time in their method, such that the integral over the response time (the one w.r.t. $u$) reduces to a simple exponential function. If a different model for the response time is used (e.g., the suggested non-parametric), I wonder how the integral can be solved and the algorithm trained with reasonable computational complexity.
	b. In the text before equation (2), I think it should be "$T=t$" and not "$T=t>d$". If we assume that $t>d$, then $P(\tilde{Y}^t=0 \mid X=x, W=w, Y=1) = 0)$. I don't think this is what the authors intended.
	c. What is the function class in the IPM?
	d. Since the authors simulate the outcomes and response times on the AIDDS, JOBS and TWINS datasets, I would suggest to call these "semi-synthetic" instead of "real-word" datasets.
	e. A few of the cited arXiv papers have already been published ((Nagpal et al., 2022), (Nie et al., 2021), (Alaa et al., 2017)).

**Questions:**

- What is the relevance of the current work in practice? Given that medical algorithms are commonly not trained online, what is the relevance of the added time dimension when learning the method? With other words, why can't you remove the time dimension, use the eventual outcome (instead of the artificially simulated observed outcome), and use CFR instead of CFR-DF?
- How does the method compare to traditional censoring approaches in the literature? In particular, why cannot a model from this literature stream be used, where "survival time" is interpreted as "time until feedback is positve"?
- Could you clarify why time independence is a reasonable assumption?

---

### Official Review · Reviewer_X2NB · 2023-10-24

**Soundness:** 3 good
**Presentation:** 3 good
**Contribution:** 1 poor
**Rating:** 3
**Confidence:** 4

**Summary:**

The paper explores heterogenous treatment effect estimation in cases where delayed responses may not be observed at analysis time. The authors extend the counterfactual regression (CFR) framework to this scenario and demonstrate the effectiveness of the proposed model though various experiments on synthetic and semi-synthetic data.

**Strengths:**

Causal analysis of delayed responses is an important area of study, and extending the CFR model to these models has a lot of potential. The paper is technically correct, and the experiments prove that the proposed fitting procedure can recover the true causal effect in synthetic data.

**Weaknesses:**

**Novelty**: The proposed model of delayed responses is well known in the literature under the name Mixture Cure Model, and it belongs to the broader class of Cure Models (see [1] for an overview). Much of sections 3 and 4 are already well established:

*	Equation (4) is the definition of a mixture cure model (see equation 1 in [1]).

*	Similar EM algorithms are commonly used for these models, see for example [2] and [3].

*	Identifiability results for similar cure models are well established, see [4] and [5].

The primary novelty appears to be the application of the CFR to cure models. This is potentially a valuable contribution but is not how the paper is currently framed. Furthermore, a significant amount of related work is omitted (see, for example, many of the references cited in [1])

**Experiments**: The data generating process for the synthetic data closely matches the DGP for CFR-DF but none of the baseline comparison models, so superior performance is expected by construction. While useful as a confirmatory study for your implementation, the comparison to these models is misleading. Instead, comparison to models that assume delayed feedback, such as [6], [7], or a standard Cox PH or AFT linear cure model, would be more convincing of the efficacy of the proposed model.


References:
> [1] Mailis Amico, Ingrid Van Keilegom. Cure Models in Survival Analysis. *Annual Review of Statistics and Its Application*. 2018.
>
> [2] Yingwei Peng, Keith Dear. A nonparametric mixture model for cure rate estimation. *Biometrics*. 2000.
>
> [3] Judy Sy, Jeremy Taylor. Estimation in a Cox Proportional Hazards Cure Model. *Biometrics*. 2000.
>
> [4] Chin-Shang Li, Jeremy Taylor, Judy Sy. Identifiability of cure models. *Statistics & Probability Letters*. 2001.
>
> [5] Leonid Hanin, Li-Shan Huang. Identifiability of cure models revisited. *Journal of Multivariate Analysis*. 2014.
>
> [6] Matthew Engelhard, Ricardo Henao. Disentangling Whether from When in a Neural Mixture Cure Model for Failure Time Data. *AISTATS*. 2022.
>
> [7] Rongqian Sun, Xinyuan Song. A Tree-based Bayesian Accelerated Failure Time Cure Model for Estimating Heterogeneous Treatment Effect. *Bayesian Analysis*. 2023

**Questions:**

In the synthetic data experiment, response times are distributed according to an exponential distribution. The memoryless property is often a poor assumption in practice, so have you explored how robust the proposed model is to event times with non-constant hazard functions?

---

### Official Review · Reviewer_bGea · 2023-11-01

**Soundness:** 2 fair
**Presentation:** 2 fair
**Contribution:** 1 poor
**Rating:** 3
**Confidence:** 4

**Summary:**

- This paper demonstrates that the parameters of interest, specifically the conditional average treatment effect on potential eventual outcomes and potential response times, are identifiable in a delayed response scenario. Furthermore, it introduces a method to estimate these parameters using a modified EM algorithm.

**Strengths:**

- This paper addresses the estimation of Heterogeneous Treatment Effects (HTE) in the context of delayed responses, which is a potentially more practical scenario than existing settings.
- They introduce two meaningful causal parameters of interest and show that these parameters are identifiable under appropriate identifiability assumptions.

**Weaknesses:**

- $\textbf{The proposed model is almost the same as the cure model in survival analysis. The author should clearly explain what are new contributions.}$

- In the real-world data analysis, the paper relies only on covariates from that dataset, which might not yield meaningful results. It would be more appropriate to generate data only for the variables that contain missing values (e.g., the missing part of potential response times, potential observed outcomes, potential eventual outcomes).

- Compared to existing HTE estimation methods, obtaining suitable data for the application of this method is expected to be challenging. It is because data need to satisfy following conditions.
	(1) Data should include the time treatment is given
	(2) The outcome at that point should be zero
	(3) Data should include the time the status changed.

- While the EM algorithm equations appear correct, it would be helpful to include a detailed derivation of the expected log-likelihood in the appendix.

- While $\tilde{y}_i^t$  may represent the observed outcome, $\tilde{Y}_i^t$ may represent $\tilde{Y}_i^t | (T=t)$ which could cause confusion.
- Expression $P(D(0)=d|X=x, Y(0)=1)$ should be modified to have the pdf form.
- An explanation of why PEHE and $\epsilon_{ATE}$ were chosen as performance measures in the experiments or references to related literature is recommended.
- The description of notation and setup should be refined. For example, (1) outcome should be zero when the treatment is given for all unit. (2) The covariate does not change over time.
- The paper's problem setting appears to differ significantly from the typical causal inference scenario. Therefore, using status instead of outcome may be more appropriate.
- Given the differences in the setup, an description of how estimation was performed for other methods may be needed.
- Since the assumptions seem stronger compared to existing causal inference method, it would be nice to discuss how to check these assumptions.

**Questions:**

- In comparison to existing methods for analyzing time-to-event data in causal inference, what do you consider the main advantages of this approach?
- In your experiments, did you utilize response time D and observed time T to estimate other methods as well? If so, how was this done?
- Did you consider the estimation of treatment effects at finite time points t?

---

### Official Review · Reviewer_5txA · 2023-11-05

**Soundness:** 3 good
**Presentation:** 3 good
**Contribution:** 3 good
**Rating:** 6
**Confidence:** 3

**Summary:**

The paper proposes to estimate the heterogeneous treatment effects (HTE) that are observed with delay. The potential response time is considered in the process of estimating HTE. Theoretical analyses show the identifiability and empirical results show improved performance against baseline methods for delayed effects.

**Strengths:**

- The paper is overall well written and I find it easy to follow the main idea and theoretical development.
- The paper identifies a very important problem of delayed effect, which is not well addressed by existing studies.
- The theoretical analysis looks sound and the identifiability is formally established, which provides some theoretical guarantee for the learning algorithm proposed.
- The learning algorithm looks solid and is described in detail. The empirical experiments show improved results against baseline methods.

**Weaknesses:**

- The empirical evaluations include only relatively small datasets. Although they are useful for benchmarking, it would be better to see how the model scales to larger datasets with more variables.
- The monotonicity assumption may hold for some applications, but not always. It would be great to see some analysis of the identifiability when this assumption does not hold.
- Figure 3 in the supplemental materials provides important insights into the proposed architecture and it would make the presentation much clearer to move it to the main text.

**Questions:**

- In Table 1, individuals are divided into four strata, where one of them is "harmful treatment" which implies that the treatment considered could be harmful to some individuals. However, Assumption 4 implies that the treatment considered would not be harmful to all individuals. Could the authors provide more clarification on this conflicting?

---

### Meta-Review · Area_Chair_w85F · 2023-12-12

**Metareview:**

The paper and its contributions need to be more correctly situated in the literature as it seems the main model is a well-known model, but not cited as such, so that the novelty can be elaborated.

**Justification For Why Not Higher Score:**

All reviewers were negative

**Justification For Why Not Lower Score:**

N/A

---

### Decision · Program_Chairs · 2024-01-16

Reject